# Dual-Energy Computed Tomography (DECT) for Diagnosing Contrast-Induced Encephalopathy (CIE) Mimicking Intracranial Hemorrhage (ICH): A Rare Case

**DOI:** 10.3390/diagnostics15192426

**Published:** 2025-09-23

**Authors:** Yuhong Shen, Tianhe Ye

**Affiliations:** 1Department of Radiology, Union Hospital, Tongji Medical College, Huazhong University of Science and Technology, Wuhan 430022, China; 13980594821@163.com; 2Hubei Province Key Laboratory of Molecular Imaging, Wuhan 430022, China

**Keywords:** contrast-induced encephalopathy, dual-energy, computed tomography, percutaneous coronary intervention complications

## Abstract

Contrast-induced encephalopathy (CIE) is a rare complication after percutaneous coronary intervention (PCI) that mimics intracranial hemorrhage (ICH). Its computed tomography (CT) findings (cortical contrast enhancement, sulci effacement) overlap with cerebrovascular conditions (e.g., cerebral infarction, subarachnoid hemorrhage). Dual-energy CT (DECT) differentiates blood/calcification from iodinated contrast medium (CM) extravasation via material decomposition, contributing to the accurate diagnosis of CIE. We report a CIE case highlighting DECT’s value. A 74-year-old woman underwent PCI. 50 min post-PCI, she had moderate headache (Numeric Rating Scale 4), dizziness, non-projectile vomiting (no seizures); vital signs were stable, no focal deficits, mannitol ineffective. Non-contrast CT demonstrated a left parietal 75 Hounsfield unit (HU) high-attenuation lesion, indistinguishable from acute intracerebral hemorrhage. Conventional non-contrast CT revealed a high-attenuation lesion (75 HU) in the left parietal lobe—indistinguishable from ICH. DECT clarified the diagnosis: virtual non-contrast maps showed CM extravasation, iodine concentration maps confirmed focal CM accumulation, and effective atomic number maps improved lesion visualization. The patient’s headache resolved within 5 h; follow-up non-contrast CT at 24 h showed complete disappearance of the lesion. She resumed clopidogrel, discharged day 3 without sequelae. This case underscores DECT’s role in distinguishing CIE (transient CM, normal neuro exam) from ICH (persistent hemorrhage), guiding safe post-PCI antiplatelet therapy.

**Figure 1 diagnostics-15-02426-f001:**
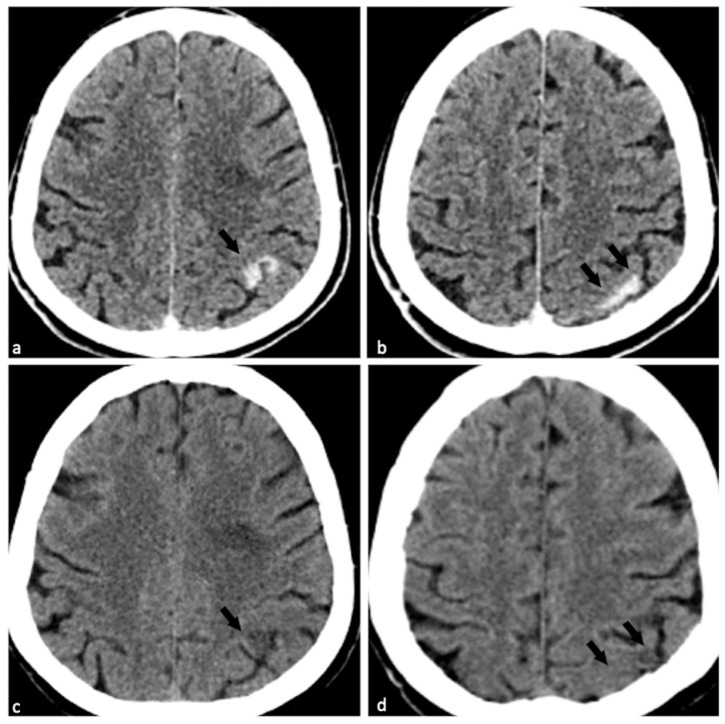
Clinical context and initial imaging findings of contrast-induced encephalopathy (CIE) mimicking intracranial hemorrhage (ICH). Conventional non-contrast computed tomography (CT), with lesions marked as “Lesion 1” (**a**) and “Lesion 2” (**b**), respectively, showed strip-like high attenuation (75 Hounsfield unit [HU]) along the left parietal gyrus ((**a**,**b**), black arrows) accompanied by mild localized brain swelling; this finding was indistinguishable from that of ICH. Virtual non-contrast (VNC) maps ((**c**,**d**), which removes iodine contrast) revealed significant contrast medium (CM) extravasation and mild low attenuation ((**c**,**d**), black arrows), confirming that the lesions were due to CM accumulation (rather than ICH). A 74-year-old woman with hypertension and diabetes was admitted for intermittent headache and chest discomfort (4 days). On admission: temperature 36.7 °C, pulse 86 bpm, respiratory rate 20 bpm, blood pressure 120/63 mmHg, BMI 28.76; conscious, negative neurological exam. Electrocardiogram showed acute inferior/lateral myocardial injury and ischemic cardiomyopathy. She received indobufen (400 mg) + clopidogrel (75 mg), then underwent percutaneous coronary intervention (PCI) (2 drug-eluting stents, 250 mL Iohexol [non-ionic low-osmolar iodine CM], 5000 IU intra-arterial. Heparin; 80 min, uneventful). The entire procedure lasted eighty minutes and was completed without any complications. Fifty minutes post-PCI, she developed moderate headache (Numeric Rating Scale 4) with dizziness and non-projectile vomiting (no seizures). Neurological exam showed bilateral equal pupils (normal light reflex) and no focal deficits; vital signs were stable (Blood Pressure 118/82 mmHg, Heart Rate 69 bpm, Respiratory Rate 20 bpm, Pulse Oximetry 98%), and she remained conscious. Administration of 125 mL mannitol failed to relieve the patient’s headache. Subsequently, the patient underwent an emergency dual-energy CT (DECT) scan. CT imaging was performed on a Siemens SOMATOM Force scanner using the FAST DE mode (DE Comp. = 0.5). Reconstruction parameters included a 1.0 mm slice thickness, Hr40 kernel, axial reconstruction, a field of view (FoV) of 200 mm, 1.0 mm increment, advanced iterative reconstruction (ADMIRE) with strength = 3, and a reconstruction region of “Head-Head-no tilt”. VNC maps, iodine concentration (IC) maps, and effective atomic number (Zeff) maps were generated using syngo.via software (version VB20A_HF08).

**Figure 2 diagnostics-15-02426-f002:**
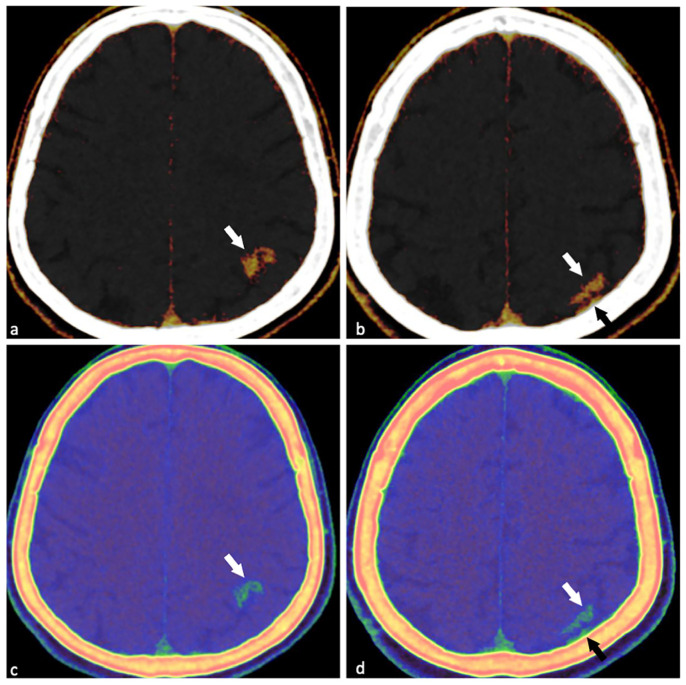
DECT findings confirming CIE. IC maps (color-coded, (**a**,**b**)): Clear CM accumulation in the left parietal lesion ((**a**,**b**), white arrows); also identifies small CM staining in the adjacent subarachnoid space ((**b**), black arrow) (indistinguishable from skull artifacts on conventional CT). Zeff maps (**c**,**d**): Shows obvious attenuation difference between normal brain tissue and skull ((**c**,**d**), white arrows); enhances visualization of the lesion adjacent to the skull ((**d**), black arrow). Appendix A summarizes the DECT quantitative metrics distinguishing lesion tissue from contralateral normal parenchyma, highlighting group-wise differences in VNC, IC, and Zeff maps. Based on the patient’s symptoms and the results of the DECT, we concluded that it was consistent with CIE. The patient was immediately transferred to the intensive care unit at our institution due to a suspected hemorrhagic stroke, and clopidogrel was stopped. After cardiac catheterization procedures, awareness of acute ischemic stroke caused by atherosclerosis, aeroembolism, vasospasm, and intimal dissection is critical [1]. Its incidence has been reported as 0.05–0.1% for coronary artery angiography and 0.12–0.4% for PCI [2]. In contrast, CIE following cardiac catheterization is uncommon and remains poorly documented in the literature. The incidence of CIE has been reported as 0.04% [3], but the actual rate may be underestimated owing to diagnostic difficulties. Presumed CIE pathophysiology involves direct neurotoxicity from iodine-based CM, which extravasate into brain tissue and disrupt the blood–brain barrier (BBB) and endothelial function [3,4]. CM characteristics influence risk: higher volume, osmolality, or concentration—especially with ionic CMs—increase susceptibility [3,4]. Notably, non-ionic isotonic CMs may also trigger injury, potentially via arterial vasospasm, microcirculatory disturbance, or elevated intracavitary pressure from power injections [5]. These factors raise vessel wall tension, break down tight junctions, further disrupt the BBB, and induce focal fluid/CM leakage, even contributing to cerebral ischemia. In the clinical management following PCI, CIE easily leads to confusion in initial assessment due to its high clinical similarity to ICH. From the imaging perspective, the typical CT findings of CIE—including cortical contrast enhancement and sulcal effacement or obliteration—exhibit significant overlap with the imaging characteristics of common cerebrovascular disorders such as cerebral infarction and subarachnoid hemorrhage, which further increases the difficulty of early differential diagnosis [4]. DECT, with its unique material decomposition technology, can effectively differentiate between the imaging signal differences caused by hemorrhage, calcifications, and iodinated CM extravasation. This technical advantage provides crucial support for the accurate diagnosis of CIE and helps prevent misdiagnosis or missed diagnosis resulting from overlapping imaging features. In our case, the patient experienced a headache within one hour after a PCI procedure (which lasted approximately 80 min) following the administration of 250 mL of Iohexol. We used a non-contrast DECT scan for emergency head imaging following cardiac catheterization. This technique generated VNC, IC, and Zeff maps. By removing the iodine contrast agent, we confirmed that the left parietal lesion was due to the contrast agent, as indicated by its low attenuation on the VNC maps. IC and Zeff maps can effectively differentiate between iodine contrast agents and hemorrhage. The IC map clearly shows iodine distribution, while the Zeff map provides additional information about the material composition of each pixel [6].

**Figure 3 diagnostics-15-02426-f003:**
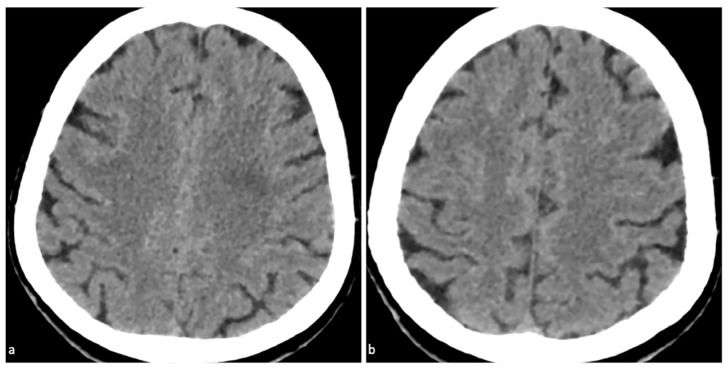
Follow-up imaging and clinical outcome of CIE. Follow-up non-contrast CT (24 h post-symptom onset). The previously seen high-attenuation lesion along the left parietal gyrus (Figure 1) has completely disappeared (**a**,**b**). Five hours post-PCI: headache mitigated; patient rested quietly. Next day: only mild dizziness. Twenty-four hours post-symptom onset: headache completely resolved. Clopidogrel was resumed after lesion disappearance; the patient was discharged on post-PCI day 3 with no neurological deficits. Most of these symptoms resolved completely within 48 to 72 h [7]. Notably, in this case, the patient developed a moderate headache accompanied by mild hypertension within one hour after PCI, which resolved within 6 h and subsided entirely within two days. Since CIE can mimic other neurological conditions, it is crucial to perform a thorough clinical examination and detailed neurological assessment for accurate diagnosis. Neurological symptoms of CIE typically occur within minutes to hours following the administration of iodine-based contrast agents [2]. Specifically, DECT findings in this case directly guided treatment decisions: VNC map confirmed CM extravasation (ruling out ICH), IC map verified focal CM accumulation, and Zeff map reinforced CM identification. This diagnosis supported continuing dual antiplatelet therapy (DAPT) with clopidogrel resumed, rather than interrupting it. By ruling out ICH, concerns about bleeding exacerbation from continued antiplatelet use were eliminated, while the risk of stent thrombosis from premature DAPT cessation was avoided.

## Data Availability

Not applicable.

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
