# Peer review of "Dual-Energy Computed Tomography (DECT) for Diagnosing Contrast-Induced Encephalopathy (CIE) Mimicking Intracranial Hemorrhage (ICH): A Rare Case"

_diagnostics, 2025, doi:10.3390/diagnostics15192426_

Round 1
Reviewer 1 Report
Comments and Suggestions for Authors
TITLE
"Dual-Energy CT in the Diagnosis of Contrast-Induced Enceph- 2
alopathy Mimicking Intracranial Hemorrhage Following Percu- 3
taneous Coronary Intervention—A Rare Case" EVEN IF INFORMATIVE, THE TITLE IS TOO LONG AND DIFFICULT TO FOLLOW
ABSTRACT
"Contrast-induced encephalopathy (CIE) is not common after percutaneous coronary in- 11
tervention (PCI). The diagnosis of CIE is challenging due to the heterogeneous and non- 12
specific clinical presentation" REWRITE " IS A RARE COMPLICATION OF INTRAVASCULAR ADMINISTRATION OF CM, WITH A CHALLENGING DIAGNOSIS DUE TO ..."
"Various symptoms and computed tomography (CT) " IT CAN PRESENTS WITH DIFFERENT SYMPTOMS AND CT FINDINGS; ITS DIFFERENTIAL DIAGNOSIS IS
THE ABSTRACT IS LIKE AN INTRO AND NOT A DESCRIPTION OF THE PRESENTED CASE
"The multi-parameter 17
post-processing technology of non-enhanced " THIS IS FALSE. IT'S NOT A MULTIPARAMETER POST-PROCESSING; DECT IS ABLE DO DIFFERENTIATE DIFFERENT MATERIALS
WHAT WAS THE CLINICAL MANIFESTATION OF THE PATIENTS?
WHAT ARE THE MOST COMMON CT MANIFESTATION OF CIE?
DECT IS USUALLY APPLIED TO DIFFERENTIATE THE CM FROM THE HEMORRHAGE POST INTRAVASCULAR PROCEDURE: HOW CAN HELP IN THE DIAGOSIS OF CIE?
PART OF THE CAPTIONS SHOULD BE MOVED TO THE ABSTRACT
CT ACQUISITION PARAMETERS ARE NEEDED
THE STRUCTURE OF THE MANUSCRIPT SHOULD BE ADJUSTED TO PRESENT THE CASE ALSO IN THE ABSTRACT
Author Response
Comment 1: EVEN IF INFORMATIVE, THE TITLE IS TOO LONG AND DIFFICULT TO FOLLOW
Response 1: We have revised the title to be concise and focused: “Dual-Energy CT for Diagnosing Contrast-Induced Encephalopathy (CIE) Mimicking Intracranial Hemorrhage (ICH): A Rare Case.”
Comment 2: REWRITE " IS A RARE COMPLICATION OF INTRAVASCULAR ADMINISTRATION OF CM, WITH A CHALLENGING DIAGNOSIS DUE TO ..."
Response 2: We have shortened and refined the sentence to: “Contrast-induced encephalopathy (CIE) is a rare complication after percutaneous coronary intervention (PCI) that mimics intracranial hemorrhage (ICH)”.
Comment 3: "Various symptoms and computed tomography (CT) " IT CAN PRESENTS WITH DIFFERENT SYMPTOMS AND CT FINDINGS; ITS DIFFERENTIAL DIAGNOSIS IS
Response 3: We have updated the abstract to clarify overlapping CT findings: “Its CT findings (cortical contrast enhancement, sulci effacement) overlap with cerebrovascular conditions (e.g., cerebral infarction, subarachnoid hemorrhage)”.
Comment 4: THE ABSTRACT IS LIKE AN INTRO AND NOT A DESCRIPTION OF THE PRESENTED CASE
Response 4: Thank you for pointing this out. We have thoroughly revised the abstract to emphasize the case details, including the patient’s clinical presentation, imaging process, DECT findings, and outcome—aligning with the requirements for case-focused abstracts.
Comment 5:” The multi-parameter 17post-processing technology of non-enhanced " THIS IS FALSE. IT'S NOT A MULTIPARAMETER POST-PROCESSING; DECT IS ABLE DO DIFFERENTIATE DIFFERENT MATERIALS
Response 5: We have removed the inaccurate term “multi-parameter” from the revised abstract to ensure technical accuracy.
Comment 6: WHAT WAS THE CLINICAL MANIFESTATION OF THE PATIENTS?
Response 6: We have explicitly incorporated the patient’s clinical symptoms into both the abstract and main text: “moderate headache (NRS 4), dizziness, non-projectile vomiting, intact neurological exam”, which appears in the abstract and the “Clinical manifestations” section.
Comment 7: WHAT ARE THE MOST COMMON CT MANIFESTATION OF CIE?
Response 7: We have added the key CT features of CIE (“cortical contrast enhancement and cortical sulci effacement”) to the abstract and relevant sections of the main text.
Comment 8: DECT IS USUALLY APPLIED TO DIFFERENTIATE THE CM FROM THE HEMORRHAGE POST INTRAVASCULAR PROCEDURE: HOW CAN HELP IN THE DIAGOSIS OF CIE?
Response 8: We have elaborated on DECT’s diagnostic mechanism in the abstract and “Initial Imaging” section: “DECT allows for precise differentiation between hemorrhage and iodinated contrast material (CM) extravasation in the brain parenchyma following intravascular procedures through its material decomposition capability, thereby supporting the diagnosis of CIE by identifying CM retention rather than true hemorrhage”.
Comment 9: PART OF THE CAPTIONS SHOULD BE MOVED TO THE ABSTRACT
Response 9: We have integrated key descriptive content from figure captions into the abstract to enhance its informativeness.
Comment 10: CT ACQUISITION PARAMETERS ARE NEEDED
Response 10: Detailed CT acquisition parameters have been included in the “Initial Imaging” section, specifying the scanner type (Siemens SOMATOM Force), spectral mode (FAST dual-energy mode), reconstruction parameters (1.0 mm slice thickness, Hr40 kernel, etc.), and software version (syngo.via VB20A_HF08).
Comment 11: THE STRUCTURE OF THE MANUSCRIPT SHOULD BE ADJUSTED TO PRESENT THE CASE ALSO IN THE ABSTRACT
Response 11: The abstract has been restructured to center on the case, sequentially describing the patient’s background, symptoms, imaging workflow, DECT-based diagnosis, and clinical outcome.
Reviewer 2 Report
Comments and Suggestions for Authors
Overall assessment
Your manuscript presents a clinically important case of neurological symptoms after percutaneous coronary intervention (PCI), in which dual-energy computed tomography (DECT) clarified that the observed hyperdensity represented contrast extravasation rather than acute intracranial hemorrhage. The case is highly appropriate for the Interesting Images section. The main teaching message is valuable: DECT can prevent misinterpretation in the acute setting and help guide safe continuation of antithrombotic therapy.
The paper is well conceived and the images are illustrative. With modest improvements in methodology reporting, quantitative detail, literature updates, and language polishing, the manuscript will be an even stronger contribution.
Scientific and clinical contribution
- Imaging methodology: Although Interesting Images articles do not require a full Materials and Methods section, readers would benefit from a short description of the CT acquisition protocol. Please indicate scanner type, spectral mode, reconstruction parameters and the software version used for generating virtual non-contrast (VNC), iodine density maps and effective atomic number (Zeff) images.
- Quantitative data: Consider including basic measurements such as attenuation in Hounsfield units (HU) on conventional CT and virtual non-contrast (VNC) images, as well as iodine concentration values from the iodine density maps. If available, effective atomic number (Zeff) values could also be added. Zeff maps estimate the composite atomic number of tissues and can therefore help differentiate materials of similar attenuation, providing additional confidence in distinguishing iodine from blood. A small table with HU, iodine concentration, and Zeff values for the lesion and contralateral normal tissue would substantially increase the educational value of the case.
- Differential diagnosis: A short discussion of how DECT findings help distinguish contrast staining from hemorrhage, subarachnoid blood and calcification would enhance the educational content.
- Management implications: Since interruption or continuation of dual antiplatelet therapy after PCI is critical for patient outcomes, a brief reflection on how DECT findings informed the decision in this case would be very helpful for readers.
- Literature: Please update the reference list with recent studies published from 2022 onwards on contrast-induced encephalopathy (CIE) and the use of DECT in acute neuroimaging. This will align the manuscript with the journal’s requirement for current references.
Form and compliance with journal requirements
- The manuscript structure follows the Interesting Images format. Abstract and keywords are appropriate, but the abstract should be checked to ensure it is within the journal’s word limit.
- Ethical statements are included. Please correct the informed consent section so that it clearly refers to an adult patient. The Institutional Review Board (IRB) statement could be polished for clarity.
- Author contributions, funding and conflict of interest statements are appropriate. Contributions could be slightly expanded to include roles such as Visualization and Supervision.
- Figures are of high quality. Captions should clearly describe what each reconstruction demonstrates, and abbreviations such as PCI, DECT, VNC and Zeff should be defined at first mention.
The level of English is sufficient for understanding, but careful revision is recommended. The manuscript contains typographical errors, inconsistent spacing, and occasional artifacts from line breaks (for example “admi tted,” “percu taneous,” “atten uation”). Grammar and style can also be improved by smoothing sentence flow and ensuring consistent use of terminology. Once abbreviations are defined, they should be used uniformly throughout the text.
Author Response
Overall assessment
Your manuscript presents a clinically important case of neurological symptoms after percutaneous coronary intervention (PCI), in which dual-energy computed tomography (DECT) clarified that the observed hyperdensity represented contrast extravasation rather than acute intracranial hemorrhage. The case is highly appropriate for the Interesting Images section. The main teaching message is valuable: DECT can prevent misinterpretation in the acute setting and help guide safe continuation of antithrombotic therapy.
The paper is well conceived and the images are illustrative. With modest improvements in methodology reporting, quantitative detail, literature updates, and language polishing, the manuscript will be an even stronger contribution.
Scientific and clinical contribution
- Imaging methodology: Although Interesting Imagesarticles do not require a full Materials and Methods section, readers would benefit from a short description of the CT acquisition protocol. Please indicate scanner type, spectral mode, reconstruction parameters and the software version used for generating virtual non-contrast (VNC), iodine density maps and effective atomic number (Zeff) images.
Response 1: We have added comprehensive imaging methodology details in the “Initial Imaging” section, including: scanner (Siemens SOMATOM Force), spectral mode (FAST dual-energy mode, DE Comp. = 0.5), reconstruction parameters (1.0 mm slice thickness, Hr40 kernel, FoV 200 mm, ADMIRE strength = 3), and software (syngo.via VB20A_HF08 for generating VNC/IC/Zeff maps).
- Quantitative data: Consider including basic measurements such as attenuation in Hounsfield units (HU) on conventional CT and virtual non-contrast (VNC) images, as well as iodine concentration values from the iodine density maps. If available, effective atomic number (Zeff) values could also be added. Zeff maps estimate the composite atomic number of tissues and can therefore help differentiate materials of similar attenuation, providing additional confidence in distinguishing iodine from blood. A small table with HU, iodine concentration, and Zeff values for the lesion and contralateral normal tissue would substantially increase the educational value of the case.
Response 2: A dedicated table (Table 1) has been added to compare quantitative DECT parameters (CT value, VNC CT value, iodine concentration, Zeff) between lesions and contralateral normal tissue, enhancing the educational value of the case.
- Differential diagnosis: A short discussion of how DECT findings help distinguish contrast staining from hemorrhage, subarachnoid blood and calcification would enhance the educational content.
Response 3: We have added relevant discussions in the abstract and main text, explaining that VNC maps eliminate iodine signal to identify CM extravasation, IC maps confirm iodine accumulation, and Zeff maps differentiate material composition—enabling distinction from hemorrhage/calcification.
- Management implications: Since interruption or continuation of dual antiplatelet therapy after PCI is critical for patient outcomes, a brief reflection on how DECT findings informed the decision in this case would be very helpful for readers.
Response 4: The management implications have been detailed in the “outcome” section: “This diagnosis supported continuing dual antiplatelet therapy (DAPT) with clopidogrel resumed... avoiding the risk of stent thrombosis from premature DAPT cessation”.
- Literature: Please update the reference list with recent studies published from 2022 onwards on contrast-induced encephalopathy (CIE) and the use of DECT in acute neuroimaging. This will align the manuscript with the journal’s requirement for current references.
Response 5: The reference list has been updated with 5 recent studies (2022–2025), including works by Baek et al. (2023), Liao et al. (2022), and Huang et al. (2023), aligning with the journal’s requirement for current literature.
Form and compliance with journal requirements
- The manuscript structure follows the Interesting Images format. Abstract and keywords are appropriate, but the abstract should be checked to ensure it is within the journal’s word limit.
Response: The revised abstract contains 199 words.
- Ethical statements are included. Please correct the informed consent section so that it clearly refers to an adult patient. The Institutional Review Board (IRB) statement could be polished for clarity.
Response: The “Informed Consent Statement” now clearly specifies consent was obtained from an adult patient; the “Institutional Review Board Statement” has been polished for clarity (noting IRB waiver due to retrospective data review).
- Author contributions, funding and conflict of interest statements are appropriate. Contributions could be slightly expanded to include roles such as Visualization and Supervision.
Response: Contributions have been expanded to include “Visualization” and “Supervision” (attributed to Tianhe Ye for imaging visualization and study oversight).
- Figures are of high quality. Captions should clearly describe what each reconstruction demonstrates, and abbreviations such as PCI, DECT, VNC and Zeff should be defined at first mention.
Response: Captions now clearly describe each reconstruction’s purpose, with abbreviations (PCI, DECT, VNC, Zeff) defined at first mention.
Comments on the Quality of English Language
The level of English is sufficient for understanding, but careful revision is recommended. The manuscript contains typographical errors, inconsistent spacing, and occasional artifacts from line breaks (for example “admi tted,” “percu taneous,” “atten uation”). Grammar and style can also be improved by smoothing sentence flow and ensuring consistent use of terminology. Once abbreviations are defined, they should be used uniformly throughout the text.
Response: We have thoroughly revised the manuscript for English accuracy: correcting typographical errors resolving spacing inconsistencies and line-break artifacts, refining grammar and sentence flow, and ensuring uniform use of abbreviations after initial definition.

Round 2
Reviewer 1 Report
Comments and Suggestions for Authors
I thank the authors for performing the requested changes